# Single-Cell Analysis to Better Understand the Mechanisms Involved in MS

**DOI:** 10.3390/ijms232012142

**Published:** 2022-10-12

**Authors:** Emilie Dugast, Sita Shah, David-Axel Laplaud

**Affiliations:** 1Center for Research in Transplantation and Translational Immunology, Nantes Université, CHU Nantes, INSERM, UMR 1064, F-44000 Nantes, France; 2CIC Neurologie, CHU Nantes, F-44093 Nantes, France; 3Service de Neurologie, CHU Nantes, F-44093 Nantes, France

**Keywords:** single-cell, multiple sclerosis, lymphocytes, monocytes, microglia, oligodendrocytes, astrocytes

## Abstract

Multiple sclerosis is a chronic and inflammatory disease of the central nervous system. Although this disease is widely studied, many of the precise mechanisms involved are still not well known. Numerous studies currently focusing on multiple sclerosis highlight the involvement of many major immune cell subsets, such as CD4+ T cells, CD8+ T cells and more recently B cells. However, our vision of its pathology has remained too broad to allow the proper use of targeted therapeutics. This past decade, new technologies have emerged, enabling deeper research into the different cell subsets at the single-cell level both in the periphery and in the central nervous system. These technologies could allow us to identify new cell populations involved in the disease process and new therapeutic targets. In this review, we briefly introduce the major single-cell technologies currently used in studies before diving into the major findings from the multiple sclerosis research from the past 5 years. We focus on results that were obtained using single-cell technologies to study immune cells and cells from the central nervous system.

## 1. Introduction

Multiple sclerosis (MS) in an inflammatory disease characterized by a chronic inflammation of the central nervous system (CNS) leading to demyelination and axonal loss. The classification of MS as an autoimmune disease at onset is not always unanimous, as there is a debate on the presence of immune cells at the lesion site being a cause or a consequence of the CNS attack. The absence of identification of a specific antigen recognized by immune cells in MS does not support the idea of it being an autoimmune disease; conversely, the effectiveness of several immunosuppressive treatments argue in its favor, as well as the animal model of multiple sclerosis, EAE (experimental autoimmune encephalitis), which is induced by murine immunization with myelin-derived peptides or with the transfer of cells specific to myelin antigens. However, it is clearly established that the disease’s evolution is caused by an imbalance between the activation and regulation of immune cells and abnormal immune cell trafficking through the blood–brain barrier to the CNS, inducing inflammation [1]. 

Although the disease’s aetiology remains unclear, it seems that genetic inheritance largely influences the disease’s onset, as genome-wide association studies (GWAS) have identified more than 200 genetic variants of susceptibility for MS, and the studies on MS-discordant monozygotic twins highlight a genetic predisposition to develop the disease [2,3]. It should be noted that genes with a higher association with MS are involved in immunity, another argument in favor of it being an autoimmune disease. Furthermore, environmental factors such as sun exposure, smoking or viral infection influence the risk of developing the disease [4,5]. Indeed, several studies have recently highlighted a strong association between EBV infection and multiple sclerosis by demonstrating the recognition of the EBV EBNA1 antigen by clonally expanded B cells [6,7,8]. Currently, not all mechanisms are fully understood, but several immune cells coming from both adaptive and innate immunity have been described as being involved in MS pathology. 

The inflammation of MS patients’ CNS is due to various immune cell infiltrates, which are mainly composed of monocytes and macrophages derived from monocytes after entry into the tissue. Monocytes are particularly abundant in the early stages of the disease, and their presence is correlated with disease severity [9]. Macrophages, which are commonly divided in two subtypes M1 and M2, have been observed in CNS lesions of MS patients [10]. M1 cells are more likely to induce tissue damage via the secretion of pro-inflammatory molecules that allow T cell differentiation into Th1 and Th17 cells. On the contrary, M2 macrophages are rather protective and promote tissue repair thanks to the secretion of anti-inflammatory cytokines and the production of extracellular matrix molecules [11,12]. However, careful thought should be given to the M1/M2 classification, as it is not as simplistic as it seems and may encompass more of a continuous spectrum of evolution given the large variety of macrophage phenotypes and their microenvironments [13]. Following macrophages, the second cell type found at the lesion site is the CD8+ T cell group [14,15,16]. These CD8+ T cells found in the cerebrospinal fluid (CSF) and in the CNS harbor a memory phenotype, and a large proportion of them display an oligoclonal distribution, suggesting a previous antigen encounter (currently not clearly identified, even if antigens derived from myelin or EBV are suggested to be recognized in MS) [8,17]. Several studies focusing on CD8+ T cells have revealed that these cells produce deleterious cytokines such as IFNγ, IL-17 or TNFα [18], as well as the granzyme B and perforin-cytotoxic molecules [19,20]. The secretion of these molecules suggests a pathogenic role for CD8+ T cells in the CNS. Among the CD8+ T cells are MAIT cells, which have been described as deleterious in the pathology of MS; however, their implication is still controversial [21]. On the other hand, some studies have demonstrated that CD8+ T cells may also have a protective role in MS, as these cells can display regulatory functions by killing pathogenic effector CD4+ T cells [22,23,24,25]. Along with CD8+ T cells, CD4+ T cells are also found at lesion sites to a lesser extent. These cells were the first to be highlighted in MS and are quite well described, as they infiltrate the CNS and are largely observed in animal models of MS, such as EAE, which is often mediated by these cells. The first CD4+ T cells described in MS were Th1 cells, characterized by the expression of the Tbet transcription factor and the secretion of IFN-γ, a pro-inflammatory cytokine considered a marker of disease progression [26]. These cells participate in inflammation through the activation of myeloid cells, CD8+ T cells and B cells. CD4+ T cells producing the IL-17 cytokine and expressing the transcription factor RORγt are known as Th17 cells and are also well described in MS [27]. They are suggested to have strong pathogenic properties and to participate in blood–brain barrier damage, leading to lesions [28] and microglial activation in the CNS [29]. Follicular helper T (TFH) cells were also suggested to be involved in MS. These cells, usually helping B cells in antibody production, harbor a defect in their distribution and might play a detrimental role in MS [30,31,32,33]. CD4+ regulatory T cells are present to counterbalance the activity of effector cells, within the context of immune homeostasis. However, in MS patients, these regulatory T cells, although present at the same frequency, show a decrease in their suppressive activity [34,35]. Contrary to T cells, B cells are scarcely observed in tissue lesions, but some follicles were described in the meninges of MS patients, and the presence of immunoglobulin oligoclonal bands suggests a role of B cells in MS, although their role is not yet clearly established. Moreover, the large efficacy of B-cell-depleting treatments in MS strongly suggests an involvement of these cells in the pathology.

In addition to these major immune cell subsets, others are described to be involved in the pathology of MS, such as NK cells, innate lymphoid cells and dendritic cells. Moreover, other specific cells of the CNS may also be actors in this pathological process, such as microglial cells, oligodendrocytes and astrocytes. 

The wide range of immune cells described in the pathological process of multiple sclerosis makes understanding the mechanisms involved particularly complex. A few decades ago, the studies mainly focused on one cell type to decipher a part of the disease process. Nowadays, many techniques using single-cell analyses are emerging to show a much broader and at the same time much finer vision of the disease course. In this review, we plan to describe the most common single-cell technologies used to study multiple sclerosis and the key data they have uncovered regarding the different cell subtypes playing a role in the pathology. 

These single-cell technologies seem to be very useful, as they allow us to better characterize the various cells and by extension to better understand how they are involved in multiple sclerosis.

## 2. Methods: Search Strategies and Inclusion Criteria

The studies included in this review were chosen through an electronic search conducted in PubMed from 2018 to July 2022. The search terms used were “multiple sclerosis” with a combination of the following: “single-cell”, “scRNAseq”, “CyTOF”, “transcriptomics”, “T cells”, “B cells”, “NK cells”, “monocytes”, “microglia”, “astrocytes” and “oligodendrocytes”. We also added some references cited in articles identified by our search strategies.

The articles were included if they used next-generation single-cell sequencing, CyTOF or single-cell qPCR on samples from patients with MS. Only articles written in English were chosen.

The major studies that used single-cell technologies to further MS research and that we described in this review are listed in Table 1.

## 3. Single-Cell Technologies: Innovative Methods for Analysis

New single-cell technologies have revolutionized our analysis of cell populations involved in MS. While researchers were previously constrained by bulk analyses, where precision was sacrificed due to collecting data from a pool of mixed cells, it is now possible to reach a great level of precision in our analyses thanks to single-cell analyses. The cell heterogeneity must be taken into account when studying cell populations, as each cell is dynamic and somewhat unique, even within a restricted cell population. Within a sorted cell population, there may still be differences in the dynamics, surface cell markers expressed, molecules secreted, gene accessibility and expression. Being able to study these factors for each individual cell enables us to cluster cell sub-populations more accurately. 

When using single-cell RNA sequencing, tissues or organs are dissociated and cells are sorted to separate each cell individually. These cells are labelled with a unique barcode allowing for each RNA sequence to be traced back to its cell of origin. The RNA can be sequenced in different ways: either by capturing the poly-A mRNA with 3′RNAseq or by capturing the total mRNA and miRNA with 5′RNAseq. The RNA is then retrotranscribed into cDNA and sequenced. The reads obtained are aligned to reference sequences from gene banks and a count matrix is generated. The downstream analysis is performed using bioinformatic tools such as R, and with representations such t-SNE (t-distributed stochastic neighbor embedding) and UMAP (uniform manifold approximation and projection) being the most common. The cells are plotted in clusters according to their similarities in gene expression, and cells with a close transcriptomic profile will be part of the same cluster. Investigating the key gene expression in each cluster helps annotate them to identify cell populations present in the dataset.

The cellular indexing of transcriptomes and epitopes or CITEseq makes it possible to retrieve data on the expression of both the RNA and chosen cell surface markers. With antibodies targeting specific surface markers, an additional library can be created. This is especially useful for markers with known low RNA counts, and can help with clustering and identifying cell populations. While this method was described using cells in suspension, there are other techniques enabling the sequencing of cells directly on the tissue, such as spatial transcriptomics. This technique combines the sequencing of mRNA with histological staining to provide a spatial resolution for the gene expression. The DNA accessibility can also be measured in cells using ATACseq. This method can be performed at the single-cell level (scATACseq), as well as the single-nuclei level (snATACseq). 

While this technology brings a great level of precision, it also has its flaws. Due to the sheer amount of time and effort that goes into sorting and staining each cell separately, it is more difficult to analyze the same number of cells as with bulk analyses. Furthermore, some transcripts may be missed during the reverse transcription due to the limited amount of cells, resulting in a possible imbalance of the gene expression from one cell to another [53]. Genes with low expression levels can be missed. Many different tools have been developed to analyze single-cell data. It can be difficult to choose the right tool or integration method, as many are very recent and the standards evolve quickly [54]. 

Cytometry by time of flight, or CyTOF, is used to analyze the proteomes of cells at the single-cell level, by targeting proteins at the surface and within cells with antibodies labelled with rare metal isotopes. The ions are then counted with a time-of-flight spectrometer. In theory, up to 100 labelled parameters can be used at once, although most current panels include up to 60 parameters. With this method, an accurate depiction of the functional state of the immune cells involved in MS can be made. 

With CyTOF, cells can be analyzed at the single-cell level to study the population heterogeneity. Furthermore, it improves on various aspects of flow cytometry, such as avoiding cell autofluorescence and spectral overlap by using metal isotopes instead of fluorochromes. On the other hand, this technique has its limits. Cells cannot be retrieved after the assay [55]. Although minimal, there is still some spillover between parameters due to the high sensitivity to metal impurities and oxide formation [56]. There is also a lower throughput than with conventional FACS technology. The data generated at a high resolution requires a more complex analysis as well. 

The classical qPCR technology can also be used at the single-cell level. Indeed, Fluidigm^®^ offers a tool to isolate up to 800 individual cells and pre-amplify chosen genes using TaqMan^®^ probes (48 to 96 genes). The pre-amplified products are then used to perform the qPCR. Despite its rather low input, this technology has the advantage of being quite easy to realize, as well as requiring a fairly simple analysis.

Cytokine production has long been studied through the use of technologies such as the enzyme-linked immunospot (ELISPOT) assay, allowing the highly sensitive and reproducible quantification of the frequency of specific cytokine- or antibody-secreting cells. However, this method does not yield data on the phenotype of these cells, and only a few analytes may be measured at the same time. While scRNAseq and CyTOF do not measure cytokine secretion by immune cells, they are able to produce precise data on cytokine transcripts and intracellular proteins at the single-cell level. 

The RNAseq and CyTOF techniques used together make for a good combination, as they focus on gene expression as well as translated proteins.

## 4. Latest Findings in MS Cellular Immunology

### 4.1. Lymphocytes

Many immune cell types were studied using a single-cell analysis in MS. As T cells are one of the most frequent cell types known to infiltrate MS lesions in the CNS, we sought to discuss them first. New T cell populations have been identified and studied for their relevancy in the physiopathology of MS, mainly from peripheral blood and CSF.

CD8+ T cells are the predominant T cell population found within CNS lesions of MS patients. While their importance in CSF and CNS inflammation has been demonstrated many times with regards to their effector memory phenotype, clonal expansion and production of effector cytokines such as IL-17, IFNγ and TNFα, currently few studies using single-cell technologies to analyze them have been published. Single-cell transcriptomics have been used to confirm the clonal expansion of cytotoxic memory CD8+ T cells in the blood and CSF of MS patients [32,57]. To date, there has not been a study published focusing on profiling only lymphocytes in the CNS of MS patients with scRNAseq. Of note, one study was performed using this technique to compare the gene expression patterns of CSF cells of a small cohort of monozygotic twins discordant for MS. For each pair of twins, one had MS and the other appeared healthy but had MRI or subclinical neuroinflammation (SCNI). Beltràn et al. confirmed the presence of clonally expanded CD8+ T cells along with CD4+ T cells and B cells in the CSF of both MS patients and subjects with SCNI [2]. In fact, the CD8+ T cells were the cell compartment with the majority of clonal expansion, and these cells strongly resembled tissue-resident memory T cells (TRM). This clonal expansion was the most significant for MS patients but was still observed to a lesser degree in subjects with SCNI, suggesting their involvement at disease initiation. 

Compared to CD8+ T cells, the use of single-cell technologies to investigate CD4+ T cells and their role in the pathology of MS has been more prolific. There have been reports of increased frequencies of Th1 and Th17 cells, as well as granulocyte–macrophage colony-stimulating factor (GM-CSF) secreting effector T cells [58,59]. While searching for specific cytokine expression patterns in PBMCs from RRMS patients using CyTOF, Galli et al. identified an expanded T helper population expressing GM-CSF, C-X-C chemokine receptor type 4 (CXCR4) and very late antigen 4 (VLA-4). Dimethyl fumarate (DMF), which is frequently used to suppress relapses in MS patients, was shown to reduce this population, noting its relevance as a therapeutic target [40]. In a later study, Diebold et al. used single-cell profiling in order to elucidate the mode of action of DMF [39]. PBMCs were collected from MS patients before and after DMF treatment and single cells were profiled using the CyTOF method. They identified a population of CD4+ memory T helper cells expressing pro-inflammatory cytokines GM-CSF and IFNγ, as well as CXCR3, which decreased after treatment. These results were then confirmed on a validation cohort, with an added correlation of these cells to axonal damage. Once again, this same population was also found to be increased in the peripheral blood following anti-VLA4 treatment. In both these studies, pro-inflammatory T helper cells expressing GM-CSF and either CXCR3 or CXCR4 appeared as therapeutic targets for disease-modifying therapies, confirming their relevance in the physiopathology of MS. 

Another study using anti-VLA4 therapy to identify targeted cell populations was conducted by Kaufmann et al. While tracking CNS-colonizing T cells, a population of pathogenic CD161+ (lymphotoxin beta) LTB+ CD4+ T cells (T09) residing in the brain tissue of progressive MS patients was identified [44]. Using a combination of scRNAseq and spatial RNAseq, Kaufmann et al. were able to find these cells not only in the peripheral blood of RRMS patients after anti-VLA4 treatment blocking their migration, but also in white and grey matter lesions in SPMS patients. These cells are likely present at disease initiation, driving acute white matter demyelination. They appear to then become resident in the cortical brain regions, highly suggesting that this cell population plays a role in disease progression throughout the CNS. Depleting these cells before they have the opportunity to colonize the CNS presents a new therapeutic strategy. 

In a recent study involving a large cohort of monozygotic pairs of twins discordant for MS, Ingelfinger et al. investigated immune signatures of PBMCs in order to assess genetic predisposition compared to environmental factors [41]. A population of naïve helper T cells highly expressing CD25 as well as the migration markers VLA4 and CXCR4 was found to be increased in twins with MS. These cells were in a transitional state from a naïve to an effector or memory phenotype. The IL-2 pathway can induce the production of GM-CSF through STAT5 signaling, so cytokine dysregulation with high levels of IL-2 may activate these transitional Th cells and push them towards a pro-inflammatory phenotype [59]. These peripheral cells could be a precursor for fully encephalitogenic Th cells residing in CNS lesions. 

In Schafflick et al.’s study, scRNAseq was performed on PBMCs and CSF cells from RRMS patients and controls [32]. A population of CD4+ T cells with a cytotoxic phenotype (expression of GZMB, PRF1, CCL5) was found to be expanded in the CSF of MS patients, but not in the blood. It would appear that this population does not encompass those that were previously described as targeted by DMF treatment (CD4+ helper cells expressing GM-CSF and CXCR4 or CXCR3) [39,40], as in this study the CD4+ T cells did not express genes coding for CXCR4 or GM-CSF. In addition to this population, T follicular helper (TFH) cells were also enriched in the CSF of MS patients. TFH cells are required for B cell maturation [60]. Using an EAE model, they showed that TFH cells indeed enhanced the B cell enrichment in the CNS, leading them to hypothesize that TFH cells are correlated with B cell abundance in the CSF. 

Although B cells infiltrate the CNS to a lesser degree compared to T cells, they remain important in the physiopathology of MS, as proven by the efficacy of anti-CD20-depleting treatments. Their exact role in this disease, however, is still not fully understood. Studying B cell phenotypes in MS is crucial to refine B-cell-targeting therapies. Various research teams have used single-cell methods to show the clonal expansion of B cells in the CSF of MS patients [2,43,49]. 

Using mass cytometry to analyze paired blood and CSF samples from MS patients and controls, Johansson et al. identified a small novel B cell population associated with MS, expressing CD49d, CD69, CD27, CXCR3 and human leucocyte antigen DR (HLA-DR) [43]. Although this phenotype strongly resembles that of memory B cells, the authors found that these cells do not fit well into established B cell subpopulations. They express CD27, which is characteristic of plasma cells, but unlike plasma cells and plasmablasts, they also express CD20. This phenotype has been observed in tonsil plasma cells that secrete IgM; however, the MS-associated B cells found in this study were IgM-negative cells. Johansson’s team hypothesized that they have a pathogenic involvement in MS through T cell stimulation with HLA-DR or antibody or pro-inflammatory cytokine secretion. The signature for these cells was mainly found during earlier disease stages, while it was diminished during secondary progressive MS. As these cells express CD20 and CD49d, they could be responsive to current treatments targeting those molecules, provided the treatment is administered early. Another study with cells from the CSF of MS patients and controls, but this time using single-cell and bulk RNAseq, was performed by Ramesh et al. [49]. They also identified polyclonal IgM and IgG B cells expanded in the CSF of MS patients and polarized towards an inflammatory and memory phenotype. The upregulation of IL-6 in these cells suggests active cross-talks with T cells in the CNS and a pathogenic role. 

Recently, the interaction between the gut and the CNS has garnered more interest in MS research. Specifically, the gut microbiome is a factor contributing to the pathogenesis of MS. IgA is the main immunoglobulin class involved in responses to microbiota, which is produced on mucosal surfaces. Pröbstel et al. conducted an experiment to understand the role of IgA+ B cells in MS [47]. Using a combination of bulk and single-cell B cell repertoire (BCR) sequencing, they showed that in MS most of the clonally related blood-to-CSF plasma cells were IgA-producing cells. The levels of IgA in CSF were increased in MS during active inflammation. Furthermore, they proved that some IgA+ plasma cell clones present in lesion sites in the CNS originated from B cell responses to MS-specific gut bacteria, suggesting that these cells traffic from the gut to the CNS and can regulate neuroinflammation via the secretion of IL-10. 

After conducting a CyTOF experiment on PBMCs from early MS patients and controls, a population of T-bet-expressing B cells was found to be abundant in early MS patients [38]. Patients were also sorted into “aggressive” and “non-aggressive” MS according to the occurrence of relapses and radiological activity under disease-modifying treatment for at least 6 months. The T-bet-expressing B cells appeared to be increased in patients with an aggressive disease form. These cells could be recruited into the CNS of MS patients [61], but further investigation is necessary to understand their role in neuroinflammation. 

While a wealth of knowledge had been gained for T cells and B cells, little is known about the presence of NK cells in MS brain lesions. NK cells can be linked to both a detrimental or a beneficial role in the pathology of MS: CD56dim CD16+ NK cells have a cytolytic capacity compared to CD56bright and CD16low NK cells, which are immunoregulators [62]. Both subtypes of NK cells have been found to be expanded in the CSF of MS patients [32]. Rodríguez-Lorenzo et al. performed CyTOF on cells from the septum—a periventricular brain region highly exposed to CSF—of MS patients, patients with dementia and controls without neurological disease [50]. The T cells observed in these samples were the main immune cells and did not vary between patient groups. However, a population of CD56bright NK cells was increased in the MS septum. These NK cells presented a migratory profile with increased expression of CD49d, CD31 and CD54 and NK cell activation markers. Considering the immunoregulatory role of CD56bright NK cells, their presence in the septum is thought to be protective against neuroinflammation.

### 4.2. Monocytes

Myeloid cells are involved in the pathology of MS as antigen-presenting cells or as effector cells in the periphery and in the CNS. There is little information about circulating myeloid cells in MS. A recent study has shown an increase in circulating classical monocytes with pro-inflammatory M1 macrophage markers in MS (CD86, CD64, CD32), as well as regulatory M2 macrophage markers (CD206, CD209, PD-L1) [62]. These results are coherent with previous research observing activated macrophages in MS lesions that present a mixed profile of pro-inflammation and regulation [10]. 

In Ingelfinger et al.’s study on monozygotic twins discordant for MS, it was revealed that a there was a shift in myeloid cells between groups [41]. Twins with MS presented less non-classical monocytes and more of a population of inflammatory classical monocytes expressing CCR2 and CD116 (GM-CSF receptor). These markers suggest cell sensitivity towards pro-inflammatory stimulation. This difference was especially noticeable in pairs of twins where MS was untreated. Therefore, these monocytes are likely to be susceptible to disease-modifying treatments. 

In Shafflick’s study on immune cells present in the blood and CSF of MS patients, one cluster of myeloid lineage cells was noted almost exclusively in the CSF, called “Mono2” [32]. These cells expressed a unique signature containing genes from classical (CD9, CD163, EGR1 and BTG2) and non-classical monocytes (C1QA, C1QB, MAF and CSF1R/CD115), perivascular macrophages (LYVE1), microglia (TREM2, TMEM119 and GPR34) and CNS-border-associated microglia (BAMs; STAB1 and CH25H). These cells have been found in other studies, with terms ranging from “microglia-like” to “CSF microglia” [49,63,64]. However, further investigation is needed to really discriminate the cell types encompassed in this population. 

In Böttcher et al.’s 2019 immune profiling study of PBMCs in early MS with CyTOF, a decreased abundance of peripheral CD141+ CD68low myeloid DCs in patients with early MS compared to healthy controls was noted [58]. As myeloid dendritic cells (mDCs) are a key component of Treg differentiation, these results consolidate the previous studies suggesting that a lack of mDCs may be linked to Treg development impairment in MS [65].

### 4.3. Microglia

In addition to macrophages originating in the peripheral blood and migrating to the CNS, there are tissue-resident macrophages coming from the embryonic yolk sac that are always in the CNS where they self-renew, called microglia. Under healthy conditions, microglia act as antigen-presenting cells and play a cleaning role by phagocytosing various debris [66]. These cells are essential for maintaining the brain tissue’s integrity. Similar to classical macrophages, microglial cells can have opposite roles, being either detrimental or beneficial in the CNS, with the balance being highly dependent on the microenvironment. The microglial cell heterogeneity has been extensively studied, with various profiles described according to the morphology, localization and surface marker expression [67,68]. Nonetheless, it is clear that this is not sufficient to differentiate all the various subsets of microglia. Even differentiating macrophages that migrate from the periphery to the CNS and microglia can prove to be a challenge. Indeed, some of their functions are redundant, but both cells also have specific roles and they intervene at different times during the disease course [69]. This is why single-cell studies can be of major interest to better define the role of each cell.

Currently, only a few studies have been conducted at the single-cell level on human brain samples from MS patients due to difficulty of accessing the brain. A few more studies have been performed on brains from deceased control donors or in animal models to decipher the mechanism of steady-state cells. These studies in healthy conditions reveal a specific transcriptional profile of microglial cells that could be modified with aging [70], but also depending on their localization in the brain [71].

A notable study used scRNA-seq to profile microglia [65]. In this study, they isolated cortical microglia from surgically resected human brain tissue without histological evidence of the CNS’s pathology. Then, they compared these profiles to those of microglia coming from the brains of patients with histologically confirmed early active multiple sclerosis. Thus, they were able to identify four different clusters in healthy controls that are partially similar to the clusters of microglial cells observed in normal adult mice. The combination of scRNA-seq microglial data from healthy controls and MS patients allowed the identification of seven clusters of microglial cells, including three that are found only in healthy controls and composed of microglia in the homeostatic state. One cluster is composed of cells from healthy and MS brains probably in a pre-activated state, as the microglia expressed a lower level of core genes (e.g., TMEM119 and P2RY12) but elevated expression levels of several cytokines and chemokines. Two other clusters are particularly enriched in cells from MS brains, and one last type is specific only to MS brain cells. These 3 clusters harbor a decreased expression level of microglia core genes but an increased expression level of APOE (a myeloid cell activation marker) and MAFB genes. Using single-cell RNA-seq data obtained from a mouse animal model, they were able to link one cluster to immunoregulation and remyelination, while another cluster was associated with demyelination. Thus, this study emphasizes the diversity of microglia that is not only related to their spatial localization, but also depends on the context and different microglial cell subtypes, with each having a very specific role in the pathology of MS. 

Another group used single-nucleus RNA sequencing technology on frozen human brain samples with cortical and subcortical lesions or lesion-free zones from MS cases and controls [51]. Interestingly, in that study they found that in addition to an activation transcriptional profile of microglia from MS brain, these cells also presented phagocytic markers as well as markers enriched for oligodendrocytes, suggesting a phagocytosis process of these cells. This last observation was then confirmed with an in vitro culture of human microglial cells with myelin from rat brains.

Another study relied on multiplexing imaging using mass cytometry to compare different kinds of MS lesions in one patient using a panel of 15+ markers [67]. Although this technology does not isolate each single cell, this can still be considered a single-cell analysis method, with the idea of looking at several cell types at the same time, observing their morphology in situ and their specific spatial location. Indeed, with this study they were able to analyze not only T and B cells, but also macrophages and microglia. Specifically, this approach allowed them to differentiate blood-derived macrophages and resident microglia with differential expression levels of the TMEM119, CD45 and CD68 markers and a modified cell morphology according to the lesion type. In addition, proteolipid protein (PLP) staining reveals the demyelinating cells that are present at a high level in active lesions. This study also shows that activated microglial cells are not only present in active lesions but also in normal-appearing white matter. This approach, which does not use high-throughput technology, is limited in terms of the number of cells and markers studied simultaneously, but has the advantage of preserving the cell morphology. This is particularly important for the macrophage or microglial function and spatial localization, and is of major interest in the study of cells’ different roles in various lesions.

Following this study, using mass spectrometry multiplexing imaging, another team focused their work on myeloid cells from active lesions of progressive MS using CyTOF [37]. Compared to the previous study, also using mass cytometry, they lacked the specific spatial location and analysis of the cell morphology, although many more cells were analyzed at the same time. It is now clearly established that the mechanisms involved in early and progressive MS are different. with less immune cell infiltration at the lesion site [72,73] and a lack of a response to immunomodulatory treatments in PMS [74]. This study was conducted to partially elucidate the mechanisms and cells involved in the progressive form of MS, as very little is currently known. Three antibody panels were used to cover a large number of markers [75], including core proteins to identify macrophages or microglia and some other immune cells, as well as antibodies to investigate phagocytic and inflammatory phenotypes. The samples came from WM from control brains and NAWM and WM lesions from MS patient brains. The major observation made is consistent with what was observed in the study of Masuda et al. [46] in that there are fewer homeostatic microglia at the lesion site but an increase in activated cells. Indeed, they show that cells found in the lesions are rather activated with a strong phagocytic profile, which is in line with the Schirmer et al. study [51]. They suggest that these microglial cells present at the site of injury may be quite beneficial as they are able to phagocytize myelin/cellular debris and thus play a cleaning role in progressive MS. On the other hand, they also found a cluster of cells that expressed high level of TNF, known to be an immune modulator with a neuroprotective effect, that is decreased in active lesions. Once again, single-cell studies highlight the great heterogeneity of microglia with opposite roles in multiple sclerosis course. Currently, the challenge would be to sort the identified cells and carry out functional experiments in order to refine the knowledge on their role in the pathology of MS.

Reich’s group analyzed brain cells from MS patients and healthy controls. They compared cells from the edge of demyelinated WM lesions at different stages of inflammation, as well as from the core of demyelinated lesions [36]. They were able to identify several types of cells—mainly microglia, but also immune cells, lymphocytes, oligodendrocytes, oligodendrocytes progenitor cells, astrocytes, neurons and vascular cells. The microglia were subclustered into subsets, including homeostatic microglia, stressed microglia and microglia inflamed in MS (MIMS), with the last one representing 25% of the immune cells in chronic active lesions. MIMS are also divided in two subgroups regarding their functions through gene expression. The first type is called “MIMS-foamy” and has a profile dedicated to regulating the inflammatory response and phagocytosis or clearance, suggesting a role in the repair of the myelin sheath. The second subgroup, called “MIMS-iron”, has a profile that is more oriented towards a pro-inflammatory function, which could in part be the cause of developing lesions. Indeed, these cells expressed high levels of the complement component, Fc-gamma receptor and inflammatory cytokines. This study also suggested strong inter-communication between cell subsets, and especially with astrocytes. Finally, they highlighted the importance of the C1q component, which is central in microglial activation and upregulated in MIMS cells on the edge of chronic active lesions.

### 4.4. Oligodendrocytes and OPC

Oligodendrocytes (OL) are glial cells derived from OPCs (oligodendrocyte progenitor cells) and are central in the CNS, as they form the myelin sheath of nerve fibers. In multiple sclerosis, these cells are targeted by immune cells, causing their destruction and leading to axonal loss. In some conditions, a process of remyelination can be observed and is associated with neurological function improvement. This remyelinating process is dependent on OPC activation, proliferation, migration to the lesion site and differentiation into oligodendrocytes [75]. This remyelinating process is governed by highly regulated mechanisms that vary according to location and result in a significant heterogeneity of OLs, which was well shown by Marques et al. in mice [76]. Falcão et al. used a mouse model of MS and highlighted a modification in the OL signature during the disease with RNAseq [77]. Regarding multiple sclerosis, this OL heterogeneity has been analyzed by Jäkel et al. using snRNA sequencing [42]. They compared cells from control brains and different brain regions from MS patients. Combining all of the sn-RNAseq data they generated, they were able to define seven OL clusters, an OPC cluster and a cluster of committed OL precursors (COPs) with strong similarity to the cluster identified in the mouse model. Among the OL clusters, one corresponded to immune oligodendroglia (imOLG). Oligo6 is an intermediate state between OPC and mature OL, and other clusters include mature OL with Oligo1 and Oligo5 representing end states of maturation without the expression of myelin genes but rather genes of cell signaling, cell–cell adhesion and viability. In contrast, Oligo3 and Oligo4 expressed genes involved in the myelination and membrane assembly. Finally, Oligo2 expressed genes associated with the cellular stress response. They revealed changes in the OL cluster frequency in patients with MS compared to controls, although each cluster is found in both groups. Thus, they found less OPCs and Oligo6 in MS lesions (independently of the lesion type) and MS NAWM compared to controls. The Oligo1 cluster was almost entirely absent in MS tissue, while Oligo2, oligo3, Oligo5 and imOLG were enriched in MS compared to the controls. Interestingly, they showed that in MS, mature OLs expressed higher levels of genes associated with myelin, suggesting a potential myelination process. Finally, they also identified several genes differentially expressed according to lesion type; however, these results must be confirmed on a larger scale. This study highlighted the complexity of the disease and eventually gave ideas for new therapeutic strategies.

Although Schirmer et al.’s study [51] was not as detailed as Jäkel et al.’s study [42] regarding OL cells, their work also showed that OL cells harbor a profile of genes enriched in stress pathways, similar to the Oligo2 cluster that was increased in MS patients in Jäkel et al.’s study. They also observed a decreased expression of genes involved in differentiation processes and myelin synthesis. This was in accordance with the previous study where they showed a decrease in OPCs and Oligo6, subsets of cells upstream of mature OLs.

In a recent study by Kihara et al. [45], brain samples from RRMS and SPMS were compared using snRNA-seq. Their work confirmed that the pathological process is dramatically different through the disease evolution. OL cells are characterized by several genes, of which the most relevant are PLP, MOG and MAG. These genes were expressed at significantly higher levels in RRMS when compared to SPMS, which could explain the loss of OLs in SPMS. In this study, as in the previous one, they observed an increase expression of genes related to cellular stress in RRMS vs. SPMS, which could also explain the decreased OL level in SPMS. They also found a gene signature of the nonsense-mediated mRNA decay (NMD) pathway, whose role is to degrade aberrant mRNAs, perhaps explaining the accumulation of this mRNA, which could be deleterious in OLs. Finally, they also described a gene signature of OPC maturation and myelination that is greater in RRMS compared to SPMS, suggesting the existence of a repair potential in RRMS that is lost in SPMS.

Combining all of these observations, the single-cell studies demonstrate that OPCs and OLs exhibit dramatic heterogeneity, pre-existing in a steady state but accentuated during the pathology of MS, with variability dependent on both the cell location in the MS brain and the disease form.

### 4.5. Astrocytes

Astrocytes (AS) are another subtype of glial cells with a variable concentration depending on their localization in the brain. These cells have several functions, including structural support and synaptic signalling. They also facilitate neuronal calcium signaling, control neurotransmitter release and uptake and modulate the blood–brain barrier function [78]. These cells are mainly characterized by their expression of the glial fibrillary acidic protein (GFAP). This protein has been known to increase during neuroinflammation, associated with the conversion of astrocytes to a neurotoxic phenotype called A1. Furthermore, another phenotype known to be neuroprotective has been coined as A2 [79]. However, apart from this A1/A2 distinction, few studies have described the possible heterogeneity in humans and even fewer in MS. Single-cell analyses of mice models confirmed the AS heterogeneity and suggested that they present a variety of functions depending on their localisation [80].

In Schirmer et al.’s study, snRNA-seq was performed on CNS cells from cortical grey matter and adjacent subcortical WM MS lesion areas at various stages of inflammation and demyelination, as well as from control tissues from unaffected individuals [51]. They demonstrated a change in the transcriptional profile of the AS according to the different demyelination zones. An astrogliosis was revealed through high GFAP levels in regions of subcortical demyelinated WM, which was not observed in the demyelinating area of the cortex. This increase in GFAP was also associated with an increase in the CD44 marker, as well as the BCL6 and FOS transcription factors in these AS from subcortical WM, suggesting their reactivity. On the other hand, they demonstrated a decrease in the expression of genes associated with both glutamate and potassium homeostasis in AS from the cortical area. The differences observed depend on the lesion area, clearly suggesting a differential role of these cells in demyelinating lesions.

Another study using scRNA-seq aimed to analyze AS from MS and control patients [52]. To improve the relevance of the scRNA-seq data, they combined their data with other datasets [42,51,81]. Throughout this analysis, they identified an AS subset that is largely increased in MS patients and that harbors a decrease in NRF2 activation but increases in MAFG activation, GM-CSF signaling, DNA methylation and pro-inflammatory pathway activity. This observation, which was also found in the EAE animal model, was confirmed by immunostaining MS brain samples, with an increase in MAFG and a decrease in NRF2 expression in AS located in active lesions in WM. These data suggest a potential role of MAFG+ AS cells in MS.

Absinta et al. carried out a remarkable study with snRNA-seq of brain samples from the edge of demyelinated WM lesions at different stages of inflammation to the core of demyelinating lesions, as compared with WM from healthy brains [36]. They observed a clear variability in cell types across the locations when compared to a healthy control (HC). In periplaque WM from MS, one-third of AS upregulated several molecules dedicated to the unfolding protein response, oxidoreductase activity and the regulation of cell death. In chronic active lesions, an increase in AS was found. These cells are also prevalent in chronic inactive areas. The AS increased on the edge of chronic active lesions expressed molecules in response to lipids, corticosteroids and wounding, and also previously with pro-inflammatory genes associated with the A1 profile [79]. With all of their data, they were able to model a hub formed by inflamed microglia and inflamed astrocytes that exist in chronic active lesions and allow the connection with immune cells and other glial cells. Nevertheless, this hub lacks a complex network in chronic inactive lesions and in lesion cores due to a lack of interaction with the glial cells.

The recent study by Kihara et al. revealed increased GFAP and CD44 expression in RRMS vs. SPMS [48] which is also observed in the AS of the subcortical WM in the Schirmer et al. study [51]. Inversely, the AS from RRMS patients expressed lower levels of genes related to the antioxidant pathways compared to SPMS. Interestingly, they showed in RRMS an upregulation of the C3 gene, which was also observed by Absinta et al. in the “astrocytes inflamed in MS” cell subset [76], which is associated with A1-type pro-inflammatory AS, as well as PFN1, which is involved in the morphology and motility of AS. Thus, this study emphasized the differences in the AS cells through the disease form. However, the previously described A1 and A2 phenotypes do not seem to be so clear in this study with respect to the astrocytes themselves, but also with respect to the microglial cells that participate in the orientation of the differentiation of these AS; thus, this requires further investigations.

## 5. Conclusions

In recent years, single-cell technologies have become progressively popular in the study of multiple sclerosis. These technologies, although sometimes difficult to implement, offer the opportunity to study in a very targeted way the mechanisms involved in the disease process. This has allowed the identification of new cell subtypes altered throughout the disease course, both in the periphery and in the CNS, as summarized in Figure 1; thus, this opens up the field of treatment opportunities. As a large amount of new populations and cell targets are discovered with these large-scale data-generating methods, this also complexifies our current knowledge with new avenues to investigate and decipher.

## Figures and Tables

**Figure 1 ijms-23-12142-f001:**
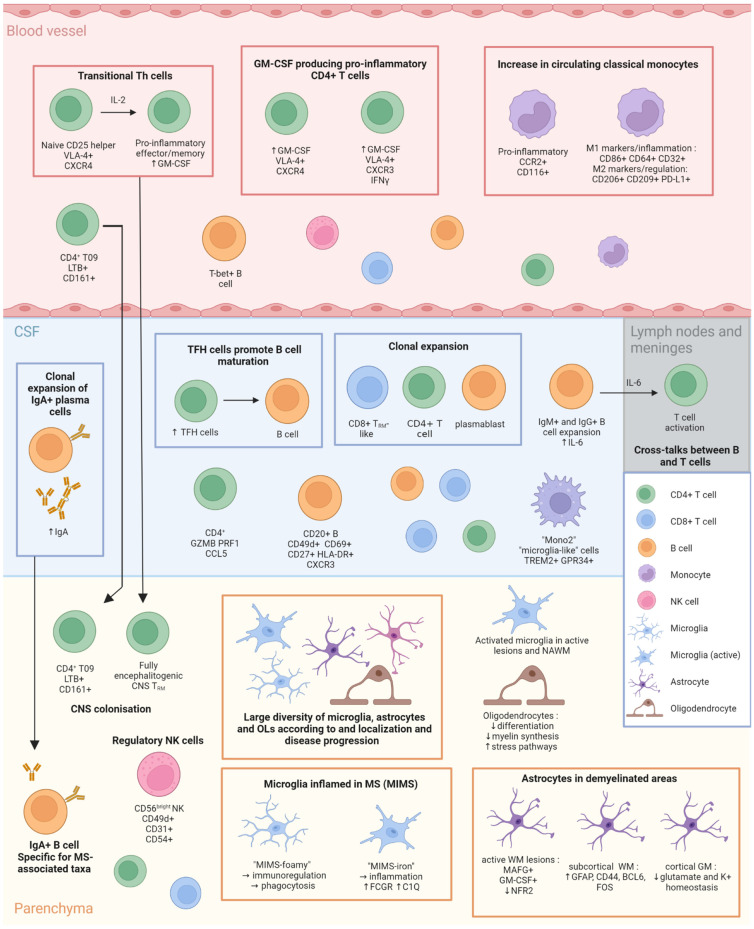
Representation of relevant cell types found in the blood, CSF and CNS during MS inflammation. Cells represent highlights discussed in this review. Figure made by authors using Biorender.com.

**Table 1 ijms-23-12142-t001:** Overview of the studies that used single-cell technologies described in this review.

Authors	Single-Cell Technology	Samples	Studied Cells	Key Findings
Absinta et al. [36]	snRNAseq	Frozen brain tissue from progressive MS patients (*n* = 5) and age- and sex-matched non-affected, non-demented controls (*n* = 3)	Microglia, immune cells, lymphocytes, oligodendrocytes, OPC, astrocytes, neurons and vascular cells	Large diversity of cells. Microglia inflamed in MS (MIMS) are divided into 2 subsets: “MIMS-foamy” and “MIMS-iron”. Strong inter-communication between cell-subsets and especially with astrocytes. Importance of the C1q component, central in microglia activation and upregulated in MIMS cells on the edge of chronic active lesions.
Beltràn et al. [2]	scRNAseq	CSF samples and PBMCs from clinically discordant monozygotic twin pairs (*n* = 4), co-twins with SCNI from whom CSF of the corresponding MS twin was not available (*n* = 4), subjects with encephalitis (*n* = 2), NIC controls with intracranial idiopathic hypertension (*n* = 2)	Immune cells in CSF, mainly lymphocytes	Clonal expansion of CD8+ T cells, plasmablasts and CD4+ T cells. Clonally expanded T cells showed tissue-resident memory phenotype.
Böttcher et al. [37]	CyTOF	WM from control brains (*n* = 8) and NAWM (*n* = 10) and WM (*n* = 10) lesions from progressive MS patient brains	Myeloid cells	Fewer homeostatic microglia at the lesion site but an increase in activated cells. Cells found in lesions are activated with a strong phagocytic profile. A cluster of cells that expresses a high level of TNF is decreased in active lesions.
Couloume et al. [38]	CyTOF	Frozen PBMCs from MS patients at first relapse (*n* = 11) and healthy controls (*n* = 8)	Myeloid and lymphoid cells	Increase in T-bet-expressing B cells and CD206+ classical monocytes in early MS. T-bet+ B cells enriched in aggressive MS patients.
Diebold et al. [39]	CyTOF	Frozen PBMCs from RRMS patients (*n* = 31) before and at 12 weeks and 48 weeks of DMF treatment	Lymphocytes	Identification of GM-CSF+ IFNγ+ CXCR3+ memory helper T cells linked to axonal damage in MS and reduced with DMF treatment.
Galli et al. [40]	CyTOF	Frozen PBMCs and CSF cells from RRMS patients (*n* = 39), healthy controls (*n* = 29) and non-inflammatory disease controls (*n* = 31)	Lymphoid and myeloid cells	Identification of GM-CSF+ VLA-4+ CXCR4+ memory helper T cells expanded in blood and enriched in CNS of MS patients. This population is reduced by DMF treatment.
Ingelfinger et al. [41]	CyTOF scRNAseq	Frozen PBMCs from monozygotic twin pairs clinically discordant for MS (*n* = 61)	Lymphoid and myeloid cells	MS patients show an increase in inflammatory classical monocytes and transitional helper T cells hyper-responsive to IL-2 with expressing migration and proliferation markers. These immune perturbations are non-heritable.
Jäkel et al. [42]	snRNAseq	WM from human controls (*n* = 5) and progressive MS patients. For MS block, different WM areas were used: NAWM (*n* = 3), active (A) (*n* = 2), chronic active (CA) (*n* = 4), chronic inactive (CI) (*n* = 3) and remyelinated (RM) (*n* = 2) lesions	Neurons, OLs, OPCs, committed OL precursors, astrocytes, vascular smooth muscle cells, pericytes, endothelial cells and immune cells	Loss of the Olig1 mature OL in MS patients. Modification of the transcriptional profile of the other OL. Depletion of Olig6 and OPC in MS and increased expression of myelin genes in mature OL.
Johansson et al. [43]	CyTOF	Frozen PBMCs and CSF cells from MS patients (*n* = 14) and controls (*n* = 25)	Lymphoid and myeloid cells	Identification of a CD49d+ CD69+ CD27+ CXCR3+ HLA-DR+ B cell population associated with MS.
Kaufmann et al. [44]	scRNAseq spatial RNAseq	Frozen PBMCs from RRMS patients during (*n* = 10) and after (*n* = 9) natalizumab treatment, RRMS (*n* = 11) and PPMS (*n* = 10) patients without immunomodulatory treatment and matched healthy controls (*n* = 31). Fresh-frozen brain tissue from progressive MS patients and controls.	Lymphoid and myeloid cells	Identification of population of pathogenic CD161+ LTB+ CD4+ T cells (T09) in peripheral blood of RRMS and PPMS patients and in CNS lesions of progressive MS patients. These cells are likely present at disease initiation and become CNS-resident in cortical brain regions.
Kihara et al. [45]	snRNAseq	Non-lesioned samples from frozen brains of RRMS (*n* = 5) and SPMS patients (*n* = 5)	Astrocytes, endothelial cells, excitatory and inhibitory neurons, OLs, OPCs, lymphocytes, myeloid cells (microglia/macrophages) and pericytes	When comparing RRMS and SPMS: lower expression of excitatory neuronal markers in RRMS. Decreased expression of OL markers in SPMS. OPC maturation and myelination gene signature is greater in RRMS. RRMS astrocytes showed upregulation of marker genes for pan-reactive astrocytes. Immediate–early genes were increased in RRMS astrocytes.
Masuda et al. [46]	scRNA-seq	Healthy brains from epileptic patients (*n* = 5); brains from RR-MS patients or patients with first manifestation of MS (*n* = 5)	Isolated microglia	Microglia associated with MS brain expressed lower level of core genes but higher level of other genes including cytokines and chemokines
Pröbstel et al. [47]	scRNAseq	Fecal samples, PBMCs and CSF cells from CIS (*n* = 4) and RRMS patients (*n* = 39) and healthy controls (*n* = 31). Snap-frozen brain tissue from MS patients (*n* = 12) (cortical and subcortical areas with acute and chronic active lesions) and controls (*n* = 5).	B cells	IgA enrichment in inflamed CNS. IgA-producing B cells specific to MS-associated gut microbiota traffic to the CNS of active MS patients.
Ramaglia et al. [48]	Multiplexing imaging by mass cytometry	Different kinds of lesions in one RR-MS patient (*n* = 1) and a non-neurological control case (*n* = 1) that died of cardiac arrest. Study performed on frozen samples	Macrophages, microglia, T cells and B cells	Modification in cell morphology regarding the lesion type; activated microglial cells are not only present in active lesions but also in the NAWM
Ramesh et al. [49]	scRNAseq	Paired PBMCs and CSF from RRMS patients (*n* = 18), non-MS neurological controls (*n* = 3) and healthy controls (*n* = 3)	Lymphoid and myeloid cells	Polyclonal IgM+ and IgG+ B cells are expanded in CSF of MS patients and polarized towards an inflammatory and memory phenotype.
Rodríguez-Lorenzo et al. [50]	CyTOF	Immune cells from fresh brain tissue of progressive MS patients (*n* = 12), patients with dementia (*n* = 8) and controls (*n* = 10). FFPE brain tissue from MS patients (choroid plexus, *n* = 10; periventricular areas, *n* = 7) and non-neurological controls (choroid plexus, *n* = 8; periventricular areas, *n* = 5). Blood and CSF were also collected.	Lymphoid and myeloid cells	CD56^bright^ NK cells with a migratory profile and NK cell activation markers are increased in MS septum. These cells may be immunoregulatory.
Schafflick et al. [32]	scRNAseq	Fresh PBMCs and CSF cells from RRMS patients (*n* = 6) and control patients (*n* = 6)	Lymphoid and myeloid cells	“Mono2” myeloid cells that express intermediate monocyte, perivascular macrophage, border-associated macrophage and microglia markers are specific to CSF of MS patients. Cytotoxic CD4+ helper T cells and Tregs expanded in CSF of MS patients.
Schirmer et al. [51]	snRNAseq; multiplex in situ hybridization	Frozen human brain samples with cortical and subcortical lesions or lesion-free zones, from MS cases (*n* = 17) and controls (*n* = 16)	Excitatory and inhibitory cortical neurons, astrocytes, OL lineage cells, microglia and smaller cell populations	Activation transcriptional profile of microglia from MS; expression of phagocytic markers. Modification of the astrocyte transcriptional profile depending on the demyelinating areas.
Wheeler et al. [52]	scRNA-seq	Fresh control (*n* = 4) and MS brains from surgery (*n* = 6)	Astrocytes	Large increase in astrocytes in MS vs. CTRL associated with a decrease in NRF2 activation but increases in MAFG activation, DNA methylation, GM-CSF signaling and pro-inflammatory pathway activity

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
