# Peer review of "Single-Cell Analysis to Better Understand the Mechanisms Involved in MS"

_ijms, 2022, doi:10.3390/ijms232012142_

Round 1
Reviewer 1 Report
The work ijms-1935492 by Dugast, Shah, and Laplaud, entitled “Studying MS at the Single Cell level to better understand the mechanisms involved” describes the use of single cell technology and its relevance in multiple sclerosis.
The topic is very interesting and the Authors provide a quite exhaustive and concise description of the field, well balancing the content of the different chapters. The summarizing Table is appropriate and valuable, also the Figure is catchy and informative.
Although the Authors collect data from an electronic search conducted in PubMed from 2018 to 2022 describing the most recent single-cell technologies, in Chapter 3 (Single-cell technologies: innovative methods for analysis), they should consider a comparison with earlier technologies and discussing, for instance, single-cell resolution enzyme-linked immunospot (ELISPOT) assay for the analysis of cytokine production in PBMC (Quantification of self-recognition in multiple sclerosis by single-cell analysis of cytokine production. Pelfrey CM, Rudick RA, Cotleur AC, Lee JC, Tary-Lehmann M, Lehmann PV. J Immunol. 2000 Aug 1;165(3):1641-51. doi: 10.4049/jimmunol.165.3.1641). This will provide a deeper perspective of the field and add value to the entire work.
Minor issues:
The title should better read: “Single cell analysis to better understand the mechanisms involved in MS”
The following references are identical: Refs. 2 and 42; 32 and 41. Please double check them all.
There are some grammar errors (for instance page 12 line 404, “which does not used”), punctuation and stylistic forms to be corrected and/or improved.
Author Response
Although the Authors collect data from an electronic search conducted in PubMed from 2018 to 2022 describing the most recent single-cell technologies, in Chapter 3 (Single-cell technologies: innovative methods for analysis), they should consider a comparison with earlier technologies and discussing, for instance, single-cell resolution enzyme-linked immunospot (ELISPOT) assay for the analysis of cytokine production in PBMC (Quantification of self-recognition in multiple sclerosis by single-cell analysis of cytokine production. Pelfrey CM, Rudick RA, Cotleur AC, Lee JC, Tary-Lehmann M, Lehmann PV. J Immunol. 2000 Aug 1;165(3):1641-51. doi: 10.4049/jimmunol.165.3.1641). This will provide a deeper perspective of the field and add value to the entire work.
We decided to focus on recent publications using newer and high-throughput single-cell technologies. However, the reviewer makes an insightful point here about older technologies. Therefore we added a section in Chapter 3 regarding methods like ELISPOT, its uses, limitations, and how some of these limitations may be overcome by scRNAseq/CyTOF.
The title should better read: “Single cell analysis to better understand the mechanisms involved in MS”
The title was changed according to this suggestion.
The following references are identical: Refs. 2 and 42; 32 and 41. Please double check them all.
References were doubled checked and corrected.
There are some grammar errors (for instance page 12 line 404, “which does not used”), punctuation and stylistic forms to be corrected and/or improved.
Grammar errors were corrected.
Reviewer 2 Report
The review article by Dugast et al. discusses briefly the major single-cell technologies currently used in research and the results that were obtained using single-cell technologies to study immune cells and cells from the central nervous system, concerning MS.The article is highly topical and well written.
Just an observation/concern: when the authors discuss the CD27+ CD20+ B cell they state that: These cells do not fit well into established B cell subpopulations. However, CD27+ CD20+ CD19+ cells are Memory B cell. Maybe they should clarify this point
Author Response
Just an observation/concern: when the authors discuss the CD27+ CD20+ B cell they state that: These cells do not fit well into established B cell subpopulations. However, CD27+ CD20+ CD19+ cells are Memory B cell. Maybe they should clarify this point.
Although CD27+ CD20+ CD19+ B cells indeed fit into the memory B cell category, this cell population ("MS associated B cells") is described by the authors themselves as not fitting into established phylogeny of B cell subtypes (precursor, naive, memory, plasma cells). They do not provide more information regarding its association (or lack thereof) with memory B cells, and are limited by their choice/number of antibodies used in CyTOF to define a more precise phenotype.
It is specified that these MS associated B cells are unlike bone marrow and circulating plasmablasts (due to CD20 expression) and unlike tonsil plasma cells (due to lack of IgM production).